# Fuzzy Synthetic Evaluation of the Critical Success Factors for the Sustainability of Public Private Partnership Projects in China

**Binchao Deng** [1,*], **Dongjie Zhou** [1], **Jiachen Zhao** [1], **Yilin Yin** [1,2] **and Xiaoyu Li** [1]

1   School of Management, Tianjin University of Technology, Tianjin 300384, China; djzhou2015@sohu.com (D.Z.); zjchen_2018@sina.com (J.Z.); yinyilin@tjut.edu.cn (Y.Y.); xiaoyuli2020@sina.com (X.L.)
2   School of Management, Tianjin University, Tianjin 300072, China
*   Correspondence: dbchao1985@tju.edu.cn

**Abstract:** Public Private Partnership (PPP) projects have attracted wide attention from academia and industry over the past 20 years, however, they have been plagued by certain factors. This study identified, classified, and evaluated the success factors that may affect PPP projects for achieving sustainability. First, a list of 32 critical success factors were categorized into 3 groups, then a questionnaire survey was conducted, with 108 responses received from experts, researchers, and PPP project managers in China. Second, using a fuzzy synthetic evaluation (FSE) method, stakeholder relationships ($A_1$–$A_{10}$), external environmental ($B_1$–$B_8$), and project management of a special purpose vehicle ($C_1$–$C_{14}$) collected data at three different factor group locations in PPP projects were used in this evaluation. The results obtained nine top factors: private sector financing capacity, government credit, government commitment or guarantee, completeness of legal framework, available financial markets, the feasibility study report and implementation, effectiveness of risk management, project investment, and cost control and revenue distribution. It was demonstrated that fuzzy synthetic evaluation techniques are quite appropriate techniques for PPP projects. The research findings should impact on policy development towards PPP and Private Finance Initiative (PFI) project governance.

**Keywords:** public private partnership; critical success factors; fuzzy synthetic evaluation; sustainability; project governance





## 1. Introduction

Public private partnership (PPP) projects have been widely used to ease pressure on government finances in China since 2013. The core principles of a PPP project include win–win cooperation, risk allocation, sustainability, and revenue sharing [1–7]. However, PPP projects have been characterized as having a long implementation period [8], large investment scale [9], complex financing structure [10], financial and investment sustainability [11–13], and diverse participants [14]. The performance of PPP projects is closely related to the interests of the public and other stakeholders. A PPP project failure can cause a significant waste of social resources and affect the government's reputation. Based on a recent literature review [15–18], critical success factors over a long-term cooperation period were identified to help public and private stakeholders control PPP project performance risks.

PPP projects need a smoothly sustainable environment. However, it is not clear whether the reality matches the ideal with respect to the cooperation between the public and the private sectors, who, together, achieve value for money (VfM), project success, and sustainability. In particular, 348 PPP projects were forced to pull out of the project management library of China public private partnerships center (CPPPC) after "Implementation opinions on promoting standardized development of cooperation between the public and private sector" (No. 10, 2019 Ministry of Finance of China), because the public sector or private sector did not provide compliance documents or other non-conforming operations that

are required by CPPPC. It is indicated that several risk factors impact PPP project success, including project demands, location, financing, legal and policy environment, taxation, design, construction and technology, operation and customer service interlinked risks, and other factors [19–22]. These risks significantly threaten PPP project success. According to the CPPPC report (http://www.cpppc.org:8086/pppcentral/map/toPPPMap.do, accessed on 30 December 2020), from 2013 to 2020, 99,930 PPP projects were finished in CPPPC, with a total capital expenditure of RMB 15,278.1 billion. This was mainly invested in more than 20 industries. Similarly, over the past 25 years, more than 6000 PPP projects have reached financial closure in developing countries [23].

Despite the great benefits of PPP, these projects have faced many problems (negative effects of risk management and risk sharing, technical capacity of the private sector, investment controls, lacking a complete legal framework, the lack of a feasible operation model, and lack of a government commitment or guarantee) and many of them failed or required renegotiation [24–28]. Many studies have investigated why PPP projects fail. These studies classified reasons for failure into the following areas: risk management and allocation [29,30], stakeholder management [31,32], feasibility of operation management [33–35], government commitment or guarantee [36–39], and completeness of legal and policy framework [36,40]. All of these areas have all been extensively explored by researchers worldwide.

The indicators above show the interest researchers have had in exploring the success factors involved in delivering PPP projects worldwide. In total, 18 Critical Success Factors (CSF) were examined using a factor analysis in the context of construction PPP and PFI projects in the United Kingdom [19,41] identified 29 reliable factors, and other studies introduced fuzzy synthetic evaluation to determine CSFs and assess the factors for particular critical risk groups [32,42,43]. Ng et al. indicated that addressing the tripartite expectations (public sector, private sector, and other stakeholders) has been indispensable in ensuring the feasibility and successfulness of PPP schemes in Hong Kong [44]. Zou et al. identified the CSFs associated with relationship management in PPP projects [45]. Another study examined stakeholder perceptions of CSFs in Nigeria [46]. Finally, Osei-Kyei et al. [15] reviewed studies on CSFs from 1990 to 2013; these indicated increased worldwide research interest in PPP projects. These research publications have provided practitioners and researchers with more insights into the critical success factors and sustainability of PPP projects. Therefore, inspired by the above literature and research, this study prioritized the factors significantly influencing PPP projects. This included applying a fuzzy synthetic evaluation analysis method to overcome the issues of interdependencies and feedback among different factor-ranking alternatives. This research also developed a checklist of CSFs for PPP, which could be adopted in the further empirical and sustainable research.

The remainder of this paper is structured as follows. Section 2 offers a brief background and identification of critical success factors for PPP projects, which uses a literature review and case study. Section 3 uses fuzzy comprehensive evaluation to analyze the data of success factors that were collected by a questionnaire survey. Then, the results are discussed in Section 4, the factors are ranked, and the top nine critical success factors are obtained. Lastly, Section 5 explains the implications, limitations, suggestions for future research, and conclusions of this paper.

## 2. The Identification of Critical Success Factors for PPP Projects

The main aim of this paper is to identify CSFs which influence the establishment of a sustainable PPP, and which will enable more efficient management of PPP processes in China. For the past few decades, a major area of PPP studies receiving significant attention from academic and managerial communities relates to critical success factors (CSF). Bing et al. [19] used a factor analysis to identify 18 potential factors most likely to affect PPP and PFI project success in the UK. They included: efficient procurement, the capacity of project implementation, government guarantees, favorable economic conditions, sustainable environment, and available financial markets. According to top tier academic

journals from 1990 to 2013, Osei-Kyei and Chan [15] identified the following factors as being very significant for PPP project success: risk allocation and sharing, strong private consortia, political support, public support, and transparent procurement.

CSFs have also been categorized and assessed in studies in different countries, including: Iran [35], the UK [47], Ghana [34,48], Greece [49], Hong Kong [44,50,51], Nigeria [52], Australia [53], Vietnam [54], Malaysia [55,56], and China [43,57]. These studies, of success factors in those countries, found that different PPP projects are associated with different critical success factors. Therefore, this study identified CSFs from literature and case studies, and obtained 14 critical success factors using a comprehensive analysis, providing support for a fuzzy comprehensive evaluation.

### 2.1. Literature Review on Critical Success Factors of PPP Projects

To comprehensively research PPP projects, "critical success factor" and "PPP project" were utilized as search keywords to identify journal papers published from 2000 to 2019 in international journals using the China National Knowledge Infrastructure (CNKI) database in China, and the Web of Science database. We obtained 279 papers after the data-cleaning process, including 186 Chinese papers and 93 international journal papers.

From the above-selected literature, similarities of the success factors for PPPs are obvious, and priority is placed on nominating perceived CSFs based on perception of public and private sector participants. A large proportion of the reviewed studies arrived at their nominated CSFs based on their mean scores or experience analysis [58–84]. Therefore, it is imperative to establish and statistic the key principal success factors in life cycle of PPP projects, their interrelationships, management principles, and contribution to successful implementation of a candidate project. The researcher read these papers to ensure that no invalid records were included. Table 1 lists 30 critical success factors from the document analysis.

**Table 1.** Summary of Literature on Success Criteria from CNKI and Web of Science.

| No. | Critical Success Factors | Authers | Sum |
|---|---|---|---|
| 1 | Effective risk management and risk sharing | Chan and Chan [58]; Chan et al. [59]; Yuan et al. [60]; Mladenovic et al. [25]; Liyanage and Villalba [49]; Dixon et al., 2005 [61]; Zhang(a) [62]; Lam and Javed [24]; Cheung et al. [63]; Meng et al. [64]; Zhang(b) [15]; Qiao et al. [65]; Zhen-Yu Zhao [66]; Robert et al. [15]; Wang et al. [56]; Hofmeister and Borchert [67]; Binquan and Tong [68]; Hongping and Sudong [69]; Jingfeng et al. [70]; Qian and Xinli [71] | 20 |
| 2 | Technical capacity of private sector | Chan and Chan [58]; Chan et al. [59]; Liu et al. [72]; Liyanage and Villalba [49]; Yuan et al. [60]; Dixon et al. [61]; Zhang(b) [15] Lam and Javed [24]; Cheung et al. [63]; Meng et al. [64]; Li et al. [73]; Qiao et al. [66]; Zhang [65]; Robert et al. [15]; Xueqing et al. [43]; Wang et al. [56]; Hofmeister and Borchert [67]; Binquan and Tong [68]; Hongping and Sudong [69]; Jingfeng et al. [70]; Qian and Xinli [71]; Chou and Pramudawardhani [17]; Osei-Kyei and Chan [16]; Keers and van Fenema [30] | 24 |
| 3 | Control of investment | Ahadzie et al. [74]; Chan and Chan [58]; Lim and Mohamed [75]; Bryde and Robinson [76]; Al-Tmeemy et al. [77]; Baccarini [78]; Cox et al. [79]; Chan et al. [59]; Yuan et al. [60]; Mladenovic et al. [25]; Liu et al. [72]; Liyanage and Villalba [49]; Dixon et al. [61]; Zhang(a) [62]; Lam and Javed [24]; Meng et al. [64]; Zhang(b) [15] Li et al. [73]; Qiao et al. [66]; Robert et al. [15]; Wang et al. [56]; Hofmeister and Borchert [67]; Jingfeng et al. [70] | 23 |
| 4 | Reasonable project cooperation period | Ahadzie et al. [74]; Chan and Chan [58]; Lim and Mohamed [75]; Al-Tmeemy et al. [77]; Baccarini [78]; Cox et al. [79]; Lai and Lam [80]; Chan et al. [59]; Yuan et al. [60]; Mladenovic et al. [25]; Liu et al. [72]; Liyanage and Villalba [49]; Bryde and Robinson [76]; Dixon et al. [61]; Zhang(a) [62]; Lam and Javed [24]; Cheung et al. [63]; Meng et al. [64]; Zhang(b) [15] Li et al. [73]; Robert et al. [15]; Jingfeng et al. [70] | 22 |
| 5 | Long-term market demand | Chan and Chan [58]; Chan et al. [59]; Yuan et al. [60]; Mladenovic et al. [25]; Liu et al. [72]; Zhang(a) [62]; Cheung et al. [63]; Meng et al. [64]; Zhang(b) [15] Li et al. [73]; Robert et al. [15]; Xueqing et al. [43]; Hongping and Sudong [69]; Jingfeng et al. [70]; Qian and Xinli [71]; Xia et al. [81] | 16 |
| 6 | Long-term relationship with cooperation between government and private sector | Chan and Chan [58]; Chan et al. [59]; Mladenovic et al. [25]; Liu et al. [72]; Liyanage and Villalba [49]; Dixon et al. [61]; Zhang(a) [62]; Lam and Javed [24]; Cheung et al. [63]; Meng et al. [64]; Zhang(b) [15] Li et al. [73]; Robert et al. [15]; Xueqing et al. [43]; Wang et al. [56]; Hofmeister and Borchert [67]; Qian and Xinli [71] | 17 |
| 7 | Financial resources for private sector | Liu et al. [72]; Qiao et al. [66]; Zhang [65]; Robert et al. [15]; Wang et al. [56]; Hofmeister and Borchert [67]; Hongping and Sudong [69]; Jingfeng et al. [70]; Qian and Xinli [71] | 9 |
| 8 | Reasonable income distribution | Chan and Chan [58]; Al-Tmeemy et al. [77]; Lai and Lam [80]; Chan et al. [59]; Yuan et al. [60]; Mladenovic et al. [25]; Liu et al. [72]; Liyanage and Villalba [49]; Dixon et al. [61]; Zhang(a) [62]; Lam and Javed [24]; Cheung et al. [63]; Meng et al. [64]; Zhang(b) [15] Li et al. [73]; Xia et al. [81] | 15 |
| 9 | Complete legal framework | Qiao et al. [66]; Zhang [65]; Robert et al. [15]; Wang et al. [56]; Hofmeister and Borchert [67]; Binquan and Tong [68]; Hongping and Sudong [69]; Qian and Xinli [71]; Xia et al. [81] | 9 |

**Table 1.** *Cont.*

| No. | Critical Success Factors | Authers | Sum |
|---|---|---|---|
| 10 | Reduced public and political protests | Chan and Chan [58]; Chan et al. [59]; Yuan et al. [60]; Mladenovic et al. [25]; Liu et al. [72]; Liyanage and Villalba [49]; Dixon et al. [61]; Zhang(a) [62]; Lam and Javed [24]; Cheung et al. [63]; Meng et al. [64]; Li et al. [73]; Robert et al. [15] | 13 |
| 11 | Feasible operating model | Chan and Chan [58]; Chan et al. [59]; Yuan et al. [60]; Mladenovic et al. [25]; Liu et al. [72]; Liyanage and Villalba [49]; Zhang(a) [62]; Lam and Javed [24]; Cheung et al. [63]; Li et al. [73]; Robert et al. [15]; Xueqing et al. [43] Osei-Kyei and Chan [15]; Ahmadabadi and Heravi [35] | 14 |
| 12 | Local economic development | Chan and Chan [58]; Chan et al. [59]; Yuan et al. [60]; Mladenovic et al. [25]; Liu et al. [72]; Liyanage and Villalba [49]; Dixon et al. [61]; Lam and Javed [24]; Cheung et al. [63]; Meng et al. [64]; Zhang(a) [62]; Li et al. [73]; Wang et al. [56] | 14 |
| 13 | Government commitment or guarantee | Qiao et al. [66]; Zhang [65]; Robert et al. [15]; Xueqing et al. [43]; Wang et al. [56]; Hofmeister and Borchert [67]; Binquan and Tong [68]; Qian and Xinli [71]; House [37]; Jiang [38]; Muhammad and Johar [82]; Ahmadabadi and Heravi [35]; Ameyaw and Chan [18]; Wang et al. [83]; Kwofie et al. [49]; Emmanuel [84]; Verhoest et al. [36] | 17 |
| 14 | Financing power for private sector | Qiao et al. [66]; Robert et al. [15]; Xueqing et al. [43]; Hongping and Sudong [69]; Jingfeng et al. [70]; Xia et al. [81] | 6 |
| 15 | Fair competition for procurement process | Robert et al. [15]; Hofmeister and Borchert [67]; Binquan and Tong [68]; Jingfeng et al. [70] | 4 |
| 16 | Purchasing procedure | Robert et al. [15]; Hofmeister and Borchert [67]; Binquan and Tong [68]; Qian and Xinli [71] | 4 |
| 17 | Reductions in litigation and arguments | Chan et al. [59]; Yuan et al. [60]; Mladenovic et al. [25]; Liu et al. [72]; Liyanage and Villalba [49]; Dixon et al. [61]; Lam and Javed [24]; Cheung et al. [63]; Meng et al. [64]; Zhang(b) [15] Li et al. [73] | 10 |
| 18 | Supervision mechanism | Wang et al. [56]; Hongping and Sudong [69]; Jingfeng et al. [70]; Qian and Xinli [71] | 4 |
| 19 | Government credit | Robert et al. [15,69]; Hofmeister and Borchert [67]; Qian and Xinli [71]; Xia et al. [81] | 4 |
| 20 | Project quality | Ahadzie et al. [74]; Chan and Chan [58]; Baccarini [78]; Cox et al. [79]; Lai and Lam [80]; Chan et al. [59]; Liyanage and Villalba [49]; Dixon et al. [61]; Jingfeng et al. [70] | 9 |
| 21 | Economic policy | Robert et al. [15]; Xueqing et al. [43]; Hofmeister and Borchert [67]; Binquan and Tong [68]; Jingfeng et al. [70]; Verhoest et al. [36]; Qian and Xinli [71] | 7 |
| 22 | Financial market | Robert et al. [15]; Xueqing et al. [43]; Hofmeister and Borchert [67]; Binquan and Tong [68]; Hongping and Sudong [69]; Qian and Xinli [71] | 6 |
| 23 | Feasibility study | Zhen-Yu Zhao [69]; Robert et al. [15]; Hofmeister and Borchert [67]; Binquan and Tong [68]; Jingfeng et al. [70]; Qian and Xinli [71] | 6 |
| 24 | Project performance assessment | Osei-Kyei and Chan [16]; Jingfeng et al. [70], Mladenovic et al. [25]; Liu et al. [72]; Liyanage and Villalba [49]; Dixon et al. [61] | 6 |

**Table 1.** *Cont.*

| No. | Critical Success Factors | Authers | Sum |
|---|---|---|---|
| 25 | Stability of project operation | Chan and Chan [58]; Lim and Mohamed [75]; Cox et al. [79] | 3 |
| 26 | Flexible pricing mechanism | Wang et al. [56]; Jingfeng et al. [70]; Qian and Xinli [71]; Xia et al. [81] | 4 |
| 27 | Feasible implemention scheme | Cheung et al. [63]; Binquan and Tong [69] | 2 |
| 28 | Public support | Jingfeng et al. [70]; Qian and Xinli [72] | 2 |
| 29 | Cost-benefit assessment | Hofmeister and Borchert [67]; Binquan and Tong [68] | 2 |
| 30 | Government approval process | Hongping and Sudong [69]; Xia et al. [81] | 2 |

*2.2. A Case Study Analysis of Critical Success Factors*

A Delphi survey was conducted on PPP projects that implemented in 2013–2018, and analyze critical success factors and their processing modes for PPP projects in mainland China.

Cases study and telephone interviews were conducted out to collect data from 40 successful and failed PPP projects in China (Table 2). As a result, 17 critical success factors were identified based on the reasons for the success or failures of these cases. The analysis showed that these critical success factors were mainly related to political influence. These include the effectiveness of risk management and risk allocation, the technical capacity of the private sector, long-term market demand, a long-term cooperative relationship, financial resources for the private sector, reasonable revenue distribution, and a complete legal framework (Table 3).

**Table 2.** Typical case studies of PPP projects in China.

| No. | Successful Case | No. | Failed Case |
|---|---|---|---|
| 1 | Beijing subway Line 4 | 1 | National Sports Complex |
| 2 | Shenzhen subway Line 4 | 2 | Taiwan North-South highway |
| 3 | Dali urban and rural garbage disposal integrated system project | 3 | Wuhan Tangshunhu Sewage Treatment Plant |
| 4 | Shanghai Xinzhuang CCHP project | 4 | Changchun Huijin Sewage Treatment Plant |
| 5 | Gu'an industrial park new urbanization project | 5 | Jinzhou Sewage Treatment Plant |
| 6 | Chengdu No. 6 waterworks | 6 | Beijing No. 10 waterworks |
| 7 | Hefei Wangxiaoying Sewage Treatment Plant | 7 | Qingdao Veolia Sewage Treatment Plant |
| 8 | Guangxi Laibin B Power Plant | 8 | Shenzhen Wutongshan Tunnel |
| 9 | Jiangxi Xiajiang water conservancy project | 9 | Guangdong Lianjiang Sino-French Water Plant |
| 10 | Guangzhou–Shenzhen Expressway | 10 | Shanghai Dachang waterworks |
| 11 | Jiuquan city district cogeneration central heating project | 11 | Jiangsu Wujiang waste incineration plant |
| 12 | Nanjing Yangtze river bridge | 12 | Shanghai Yan'an Road.(E) Tunnel |
| 13 | Shaanxi south gate water conservancy project | 13 | Yangpu Bridge |
| 14 | Chongqing Fuling-Fengdu expressway project | 14 | Fujian Quanzhou Citong Bridge |
| 15 | Shenzhen University games center project | 15 | Huangqiao power plant |
| 16 | Zhangjiajie Yangjiaxi Sewage Treatment Plant | 16 | Wuhan 3rd Yangtze River Bridge |
| 17 | Wuzhong-Jingmaiyuan waste-to-energy incineration project | 17 | Zunyi North Suburb water plant |
| 18 | Weinan natural gas utilization project | 18 | Hangzhou Bay Bridge |
| 19 | Transfer Project of Tianjin NorthWater Co. Ltd. | 19 | Nanjing 3rd Yangtze River Bridge |
| 20 | Shenzhen Shajiao B power plant | 20 | Beijing five ring highway |

Successful cases were selected from the typical cases of PPP projects in the national development and reform commission website of China (https://www.ndrc.gov.cn/xwdt/ztzl/pppzl/dxal/pppdxal/, accessed on 30 December 2020). Failure cases were selected from the typical cases in the related literature in the CNKI Database. Next, the implementation effect of all the cases listed in Table 2 were analyzed, and the study sorted and determined which factors affect project success in the actual process, as shown in Table 3. The goal was to facilitate the success of more PPP projects in the total project life cycle.

**Table 3.** Critical success factors base on case analysis.

| No. | Critical Success Factor | Successful Case | Failed Case |
|:---:|:---:|:---:|:---:|
| 1 | Effectiveness of risk management and risk allocation | 1,16 | |
| 2 | Technical capacity of the private sector | 3,19 | |
| 3 | Long-term market demand | | 18,19,20 |
| 4 | Long-term cooperative relationship | 7,9 | |
| 5 | Financial resources for the private sector | 13,10,17,20 | |
| 6 | Reasonable revenue distribution | 1,2 | |
| 7 | Complete legal framework | 3 | |
| 8 | Commitment and trust between the public and private sector | 7,9 | 6,7,12 |
| 9 | Financing capacity of the private sector | 3,9,13,15,19 | |
| 10 | Fair competitive procurement procedures | 7,9,13 | |
| 11 | Transparent procurement procedures | 7,9,13 | |
| 12 | Effective monitoring mechanism | 1,3,17 | |
| 13 | Good government credit | | 3,12,16 |
| 14 | Stable economic policy | 1,11,18 | 15,17 |
| 15 | Project Feasibility Study completed and implemented | 7 | 1,2,5 |
| 16 | Flexible pricing mechanism | 1,4,6,8,10 | 8,10,14 |
| 17 | Effective exit mechanism | 10,12,20 | 4 |

*2.3. Key Success Factors for PPP Projects*

Many factors could impact on the success of PPP projects' success, and it is possible to rank and classify the relative importance of these factors. Identifying the list of critical success factors of PPP projects is done by reviewing existing literature research results and experience summaries for typical domestic PPP projects. The effect of the factors on project success can be represented as a pyramid relationship (see Figure 1), with connections between the public sector, people, and private sector. This triangular pyramid clearly shows two analytical perspectives: horizontal and vertical relationships. For the vertical perspective, the public sector, private sector, and people have a common goal: project success. This perspective mainly embodies three aspects: project governance ensured by the public sector; project management promoted by the private sector, and satisfactory feedback by people. For the horizontal perspective, the public sector and private sector work together under a project contract, and include providers who can offer high-quality public services.

Based on a literature review, case summaries, and the triangular pyramid relationship in PPP projects, this study divided project success factors into three dimensions: relationships between stakeholders, project management in a Special Purpose Vehicle (SPV), and the external environment of a PPP project. First, the relationship between stakeholders included each participant's behavior, and the partnership and contractual relationship, including the technical ability of the social sector, government credit, and other factors. Second, the project management of a SPV is composed of technology and management factors, impacting the project success in project life cycle management. Examples of this include risk allocation in risk management, investment control, and other factors. Third, the external environmental holds uncontrollable factors that affect the implementation effect of PPP projects, such as a sound legal framework and credible economic policies. Therefore, after collection, screening, and analysis processes, the literature research and case analysis yielded a final list of 32 CSFs (named $A_i$, $B_i$, or $C_i$) and grouped as A, B, and C on Table 4.

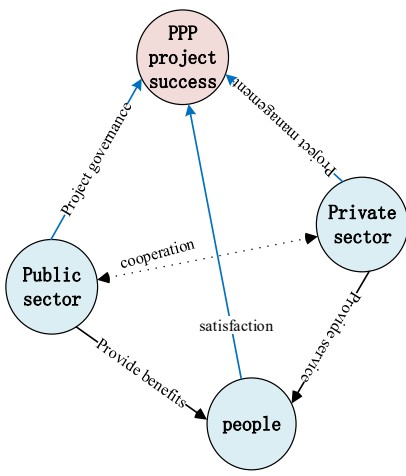

**Figure 1.** Triangular pyramid relationship in PPP projects.

**Table 4.** Critical Success Factors in a PPP project.

| Factor Group | A: Stakeholder Relationships | B: External Environmental | C: Project Management of a Special Purpose Vehicle (SPV) |
|---|---|---|---|
| Factors | $A_1$: Technical capabilities of the social sector<br>$A_2$: Government credit<br>$A_3$: Examination and approval procedure<br>$A_4$: Flexibility of pricing mechanism<br>$A_5$: Financial resources of private sector<br>$A_6$: Private sector financing capacity<br>$A_7$: Management capabilities of the private sector<br>$A_8$: Effective of the regulatory mechanism<br>$A_9$: Government commitment or guarantee<br>$A_{10}$: Long-term cooperative relationship | $B_1$: Completeness of legal framework<br>$B_2$: Public opposition and political protest<br>$B_3$: Economic policy change<br>$B_4$: Local economic development level<br>$B_5$: Available financial markets<br>$B_6$: Favorable public support<br>$B_7$: Long-term market demand<br>$B_8$: Renegotiation and arbitration | $C_1$: Feasibility study and implementation plan<br>$C_2$: Competitive bidding<br>$C_3$: Transparency of bidding<br>$C_4$: Effectiveness of risk management<br>$C_5$: Project investment and cost control<br>$C_6$: Project quality<br>$C_7$: The feasibility of operation mode<br>$C_8$: Terms of cooperation<br>$C_9$: Revenue distribution<br>$C_{10}$: Operational stability<br>$C_{11}$: Project Feasibility Study Report<br>$C_{12}$: Cost-benefit assessment<br>$C_{13}$: Performance Evaluation<br>$C_{14}$: Exit mechanism |

## 3. Methodology

Fuzzy synthetic evaluation (FSE) is a branch of fuzzy set theory [85], it has been developed and extensively applied in different disciplines to quantify multi-evaluations and multi-attributes. These fields include including knowledge management [86], human resource management [87], and construction megaprojects [88], and risk management or risk assessment in PPP projects [16,18,89]. It is an analytical tool that objectifies the subjective judgment inherent in human decision-making. Therefore, this study applies this method to construct the project success index (PSI) equation to analyze the decision-making strategies between the public sector and the private sector.

### 3.1. Questionnaire Survey

A questionnaire survey was conducted to assess the significance of the identified project success factors; it was completed by scholars, experts and project managers for different types of infrastructure-focused PPP projects. This survey allows respondents to have time to carefully ponder over their responses without any interference from researchers.

The questionnaire survey was sent by email and conducted over 6 months, with a recovery rate of 72% (108 valid questionnaires from the 150 questionnaires distributed). Following those questionnaires, the critical success factors (CSF) influencing the establishment of a sustainable PPP were extracted.

Respondents represented the private sector, financial institutions, advisory institution universities, research institutions, and the public sector. Table 5 shows the sectors and experience levels in PPP projects. A total of 49.07% of respondents were from engineering advisory institutions; 24.07% came from universities or research institutions; 18.52% came from the private sector, 2.78% came from financial institutions, and 3.7% came from other types of organizations. The distribution of respondents was consistent with PPP project stakeholders, essentially representing all stakeholders across the PPP project life cycle.

**Table 5.** The Profile of companies, respondents, and projects.

| Characteristics | Category | Number | Percentage |
|---|---|---|---|
| Sector of respondents | Private sector | 20 | 18.52 |
| | Financial institution | 3 | 2.78 |
| | Advisory institution | 53 | 49.07 |
| | Universities or research institutions | 26 | 24.07 |
| | Public sector | 2 | 1.85 |
| | Other | 4 | 3.7 |
| | Total | 108 | 100 |
| Years of working or research experience | 2 years below | 34 | 31.48 |
| | 2–5 years | 54 | 50 |
| | 6 years and above | 20 | 18.52 |
| | Total | 108 | 100 |

The data about the respondents' number of working years were as follows: 31.48% had less than 2 years of work experience; 50% had 2–5 years of experience; and the others had more than 5 years of experience. Among the 108 questionnaires managed by the respondents, 68.52% had more than 5 years of working years, with rich work experience. This screening information ensured quality, reduced the occurrence of potential risks, and improved the accuracy of the research conclusions.

Since respondents may be engaged in multiple types of PPP projects, in order to avoid the problem of overgeneralization, the author made multiple choice on the type of PPP projects the respondent has been engaged in questionnaire. The results showed that most of PPP projects engaged by respondents are distributed in the following Figure 2: such as 63 transportations, 46 water conservancy, 55 ecological construction and environmental protection, 65 municipal engineering, 42 government infrastructure construction, 29 comprehensive pipeline development, etc. This data conforms to the current development trend of PPP projects. Therefore, it is crucial to identify and analyze the key success factors of PPP projects.

The project success questionnaire included two parts: (1) the background information of respondents and their experience working on a PPP project, and (2) the Likert scale structured questions about the importance of the identified project success factors. For part one, the 108 questionnaires assessed experiences with different kinds of infrastructure types, including subways, waterworks, highways, energy, transportation, and water and waste treatment projects. This ensured the veracity and consistency of the research. For part two, respondents were requested to rate their degree of agreement against each of the identified CSFs, using a five-point Likert scale as follows: 1—Can be ignored or not important; 2—Maybe important; 3—Important; 4—Very important; 5—Most important.

Table 6 reports basic statistical parameters for the CSFs from the questionnaire, generated using SPSS24.0 software.

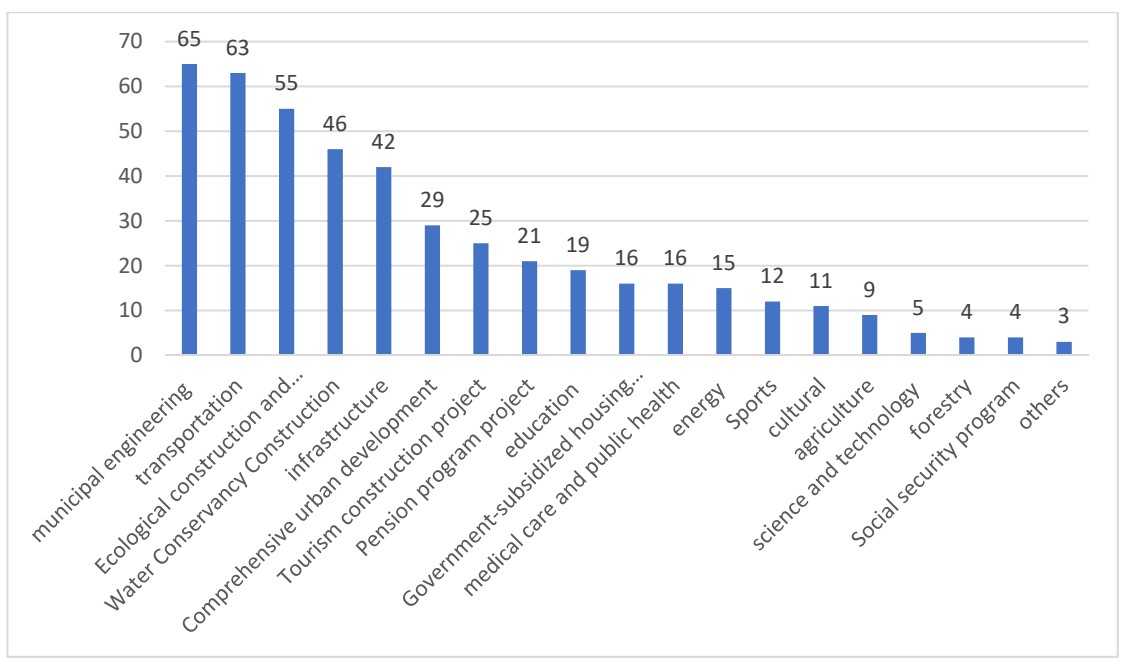

**Figure 2.** Types of PPP projects from respondents.

**Table 6.** **Statistic** of critical success factors.

| Factor Group | Factor | Mean | Standard Deviation | Normalization | Rank | Weights |
|---|---|---|---|---|---|---|
| A:Stakeholders relationship | $A_1$: Technical capabilities of the social sector | 4.2 | 0.733 | 0.5670 | 15 | 0.0989 |
| | $A_2$: Government credit | 4.58 | 0.657 | 0.9588 | 2 | 0.1078 |
| | $A_3$: Examination and approval procedure | 4.03 | 0.803 | 0.3918 | 25 | 0.0949 |
| | $A_4$: Flexibility of pricing mechanism | 3.97 | 0.703 | 0.3299 | 26 | 0.0935 |
| | $A_5$: Financial resources of private sector | 4.2 | 0.694 | 0.5670 | 16 | 0.0989 |
| | $A_6$: Private sector financing capacity | 4.62 | 0.575 | 1.0000 | 1 | 0.1088 |
| | $A_7$: Management capability of private sector | 4.25 | 0.672 | 0.6186 | 12 | 0.1 |
| | $A_8$: Effectiveness of regulatory mechanism | 4.17 | 0.755 | 0.5361 | 18 | 0.0982 |
| | $A_9$: Government commitment or guarantee | 4.37 | 0.705 | 0.7423 | 5 | 0.1029 |
| | $A_{10}$: Long-term cooperative relationship | 4.09 | 0.838 | 0.4536 | 22 | 0.0963 |

**Table 6.** *Cont.*

| Factor Group | Factor | Mean | Standard Deviation | Normalization | Rank | Weights |
|---|---|---|---|---|---|---|
| B:External environmental | $B_1$: Completeness of legal framework | 4.34 | 0.738 | 0.7113 | 6 | 0.1348 |
| | $B_2$: Public opposition and political protest | 3.9 | 0.853 | 0.2577 | 27 | 0.1211 |
| | $B_3$: Economic policy change | 4.07 | 0.732 | 0.4330 | 23 | 0.1264 |
| | $B_4$: Local economic development level | 4.04 | 0.76 | 0.4021 | 24 | 0.1255 |
| | $B_5$: Available financial markets | 4.31 | 0.703 | 0.6804 | 9 | 0.1339 |
| | $B_6$: Favorable public support | 3.65 | 0.868 | 0.0000 | 32 | 0.1134 |
| | $B_7$: Long-term market demand | 4.1 | 0.669 | 0.4639 | 21 | 0.1273 |
| | $B_8$: Renegotiation and arbitration | 3.79 | 0.724 | 0.1443 | 30 | 0.1177 |
| C:Project management of Special Purpose Vehicle (SPV) | $C_1$: Feasibility study and implementation plan | 4.42 | 0.643 | 0.7938 | 3 | 0.0881 |
| | $C_2$: Competitive bidding | 3.86 | 0.803 | 0.2165 | 29 | 0.0769 |
| | $C_3$: Transparency of bidding | 3.9 | 0.917 | 0.2577 | 28 | 0.0777 |
| | $C_4$: Effectiveness of risk management | 4.42 | 0.685 | 0.7938 | 4 | 0.0881 |
| | $C_5$: Project investment and cost control | 4.33 | 0.684 | 0.7010 | 7 | 0.0863 |
| | $C_6$: Project quality | 4.25 | 0.712 | 0.6186 | 13 | 0.0847 |
| | $C_7$: The feasibility of operating mode | 4.31 | 0.636 | 0.6804 | 10 | 0.0859 |
| | $C_8$: Terms of cooperation | 3.79 | 0.737 | 0.1443 | 31 | 0.0755 |
| | $C_9$: Revenue distribution | 4.32 | 0.609 | 0.6907 | 8 | 0.0861 |
| | $C_{10}$: Operational stability | 4.19 | 0.699 | 0.5567 | 17 | 0.0835 |
| | $C_{11}$: Project Feasibility Study Report | 4.12 | 0.758 | 0.4845 | 19 | 0.0821 |
| | $C_{12}$: Cost-benefit assessment | 4.28 | 0.681 | 0.6495 | 11 | 0.0853 |
| | $C_{13}$: Performance Evaluation | 4.21 | 0.749 | 0.5773 | 14 | 0.0881 |
| | $C_{14}$: Exit mechanism | 4.11 | 0.74 | 0.4742 | 20 | 0.0769 |

Table 6 shows that 25 critical success factors received a score at 4 or above; and 7 other factors scored between 3.65 and 4. This indicated there was some internal connection between 32 factors and project success in PPP projects. The top four scores included the financing capacity of the private sector, government credit, a feasibility study report and implementation plan, and the effectiveness of risk management, at 4.62, 4.58, 4.42, and 4.42, respectively. This indicates that respondents believe these factors have the greatest impact on PPP success projects. Therefore, PPP project participants should consider the above factors as a core concern, introducing the vitality of social capital, increasing market employment competition, improving infrastructure construction, and reducing financial pressure.

### 3.2. Data Analysis

The proposed fuzzy synthetic evaluation model is a multi-criteria evaluation model [16,43, 90] for critical success factors, requiring six steps:

Step 1: Establish the set of basic critical success factors as $U = \{f_1, f_2, \cdots f_n\}$, where $n$ is the number of critical success factors;

Step 2: Establish the grade alternatives as $L = \{L_1, L_2, \cdots L_5\}$, with the set of grade categories being the scale measurement. A 5-point Likert scale was used as the set of grade alternatives: $L_1$ is least important, $L_2$ is fairly important, $L_3$ is important, $L_4$ is very important, and $L_5$ is extremely important.

Step 3: Establish the set of basic critical success factors weight as $w = \{w_1, w_2, \cdots w_n\}$. The weighting ($w$) was determined from the survey using the following equation:

$$w_i = M_i / (\sum_{i=1}^{5} M_i), \ 0 \le w_i \le 1, \ 0 \le i \le 1,$$

In this expression, $w_i$ is weighting and $\sum_{i=1}^{5} w_i = 1$, and $M_i$ is mean score of a particular criterion or factor component.

In Step 3, the weights of each factors are calculated from the indexes obtained using SPSS. An example includes the technical capabilities for social sector ($A_1$)

$$W_{A_1} = \frac{4.2}{4.2 + 4.58 + 4.03 + 3.97 + 4.2 + 4.62 + 4.25 + 4.17 + 4.37 + 4.09} = 0.0989$$

Based on Step 3, we determine following weights of success factors (See Table 6).

Step 4: Generate a CSF evaluation matrix: $R_i = (r_{ij})_{m \times n}$, where $r_{ij}$ denotes the degree to which the alternative $L_j$ satisfies the criterion $f_i$. Let:

$$R_i = \begin{pmatrix} MF_{u_{i1}} \\ MF_{u_{i2}} \\ \cdots \\ MF_{u_{in}} \end{pmatrix} \tag{1}$$

In this expression, $MF_{u_{i1}} = \left( \frac{N_{L_1}}{N}, \frac{N_{L_2}}{N}, \cdots, \frac{N_{L_5}}{N} \right)$; $N = 108$; MF is the membership function; and $N_{L_i}$ is the number of critical success factors $f_i$ from the questionnaires. For example, when examining the first critical success factor about technical capacity in the private sector, one person selected $L_1$ as the least important; no one selected $L_2$ as fairly important; 14 people selected $L_3$ as important; 54 people selected $L_4$ as very important; and 39 people selected $L_5$ as extremely important. This resulted in the following expression:

$$MF_{u_{11}} = (\frac{1}{108}, \frac{0}{108}, \frac{14}{108}, \frac{54}{108}, \frac{39}{108}) = (0.009, 0.000, 0.129, 0.500, 0.362)$$

Step 5: Calculate the data for the weights and evaluation results, shown in Table 7.

Step 6: Generate final fuzzy synthetic evaluation results for the evaluation by considering the weighting vector and the fuzzy evaluation matrix, using the following equation:

$$T = W \times R = (w_1, w_2, \cdots, w_n) \times \begin{pmatrix} r_{11} & r_{12} & \cdots & r_{1m} \\ r_{21} & r_{22} & \cdots & r_{1m} \\ \vdots & \vdots & \vdots & \vdots \\ r_{n1} & r_{n2} & \cdots & r_{nm} \end{pmatrix} = (t_1, t_2, \cdots, t_n) \tag{2}$$

**Table 7.** Fuzzy relational matrix data indicators for critical success factors.

| | Stakeholders Relationship | Weight | | | | | Evaluation Result | | | | |
|---|---|---|---|---|---|---|---|---|---|---|---|
| 1 | $A_1$: Technical capabilities of the social sector | 0.009 | 0 | 0.13 | 0.5 | 0.36 | 0.005 | 0.008 | 0.13 | 0.44 | 0.42 |
| 2 | $A_2$: Government credit | 0.009 | 0 | 0.037 | 0.31 | 0.65 | —— | —— | —— | —— | —— |
| 3 | $A_3$: Examination and approval procedure | 0.009 | 0.019 | 0.194 | 0.49 | 0.29 | —— | —— | —— | —— | —— |
| 4 | $A_4$: Flexibility of pricing mechanism | 0.009 | 0.009 | 0.176 | 0.61 | 0.19 | —— | —— | —— | —— | —— |
| 5 | $A_5$: Financial resources of private sector | 0 | 0.009 | 0.13 | 0.51 | 0.35 | —— | —— | —— | —— | —— |
| 6 | $A_6$: Private sector financing capacity | 0 | 0 | 0.046 | 0.29 | 0.67 | —— | —— | —— | —— | —— |
| 7 | $A_7$: Management capability of the private sector | 0 | 0 | 0.13 | 0.49 | 0.38 | —— | —— | —— | —— | —— |
| 8 | $A_8$: Effectiveness of regulatory mechanism | 0 | 0.019 | 0.157 | 0.46 | 0.36 | —— | —— | —— | —— | —— |
| 9 | $A_9$: Government commitment or guarantee | 0 | 0.009 | 0.102 | 0.40 | 0.49 | —— | —— | —— | —— | —— |
| 10 | $A1_0$: Long-term cooperative relationship | 0.009 | 0.019 | 0.194 | 0.43 | 0.35 | —— | —— | —— | —— | —— |
| | External Environmental | Weight | | | | | 0.003 | 0.021 | 0.21 | 0.47 | 0.30 |
| 11 | $B_1$: Completeness of legal framework | 0 | 0.009 | 0.13 | 0.37 | 0.49 | —— | —— | —— | —— | —— |
| 12 | $B_2$: Public opposition and political protest | 0.009 | 0.037 | 0.25 | 0.45 | 0.25 | —— | —— | —— | —— | —— |
| 13 | $B_3$: Economic policy change | 0 | 0.019 | 0.176 | 0.52 | 0.29 | —— | —— | —— | —— | —— |
| 14 | $B_4$: Local economic development level | 0 | 0.019 | 0.213 | 0.48 | 0.29 | —— | —— | —— | —— | —— |
| 15 | $B_5$: Available financial markets | 0 | 0.009 | 0.111 | 0.44 | 0.44 | —— | —— | —— | —— | —— |
| 16 | $B_6$: Favorable public support | 0.019 | 0.046 | 0.361 | 0.42 | 0.16 | —— | —— | —— | —— | —— |
| 17 | $B_7$: Long-term market demand | 0 | 0.009 | 0.148 | 0.57 | 0.27 | —— | —— | —— | —— | —— |
| 18 | $B_8$: Renegotiation and arbitration | 0 | 0.028 | 0.306 | 0.52 | 0.15 | —— | —— | —— | —— | —— |
| | Project Management of Special Purpose Vehicle (SPV) | Weight | | | | | 0.003 | 0.01 | 0.15 | 0.47 | 0.37 |
| 19 | $C_1$: Feasibility study and implementation plan | 0 | 0 | 0.083 | 0.417 | 0.5 | —— | —— | —— | —— | —— |
| 20 | $C_2$: Competitive bidding | 0 | 0.019 | 0.343 | 0.4 | 0.24 | —— | —— | —— | —— | —— |
| 21 | $C_3$: Transparency of bidding | 0.028 | 0.019 | 0.25 | 0.435 | 0.269 | —— | —— | —— | —— | —— |
| 22 | $C_4$: Effectiveness of risk management | 0 | 0.009 | 0.083 | 0.389 | 0.519 | —— | —— | —— | —— | —— |

**Table 7.** *Cont.*

| | Stakeholders Relationship | Weight | | | | | Evaluation Result | | | | |
|---|---|---|---|---|---|---|---|---|---|---|---|
| 23 | $C_5$: Project investment and cost control | 0 | 0.009 | 0.093 | 0.453 | 0.444 | —— | —— | —— | —— | —— |
| 24 | $C_6$: Project quality | 0 | 0.009 | 0.13 | 0.463 | 0.398 | —— | —— | —— | —— | —— |
| 25 | $C_7$: The feasibility of operating mode | 0 | 0 | 0.093 | 0.5 | 0.407 | —— | —— | —— | —— | —— |
| 26 | $C_8$: Terms of cooperation | 0.009 | 0.019 | 0.29 | 0.546 | 0.139 | —— | —— | —— | —— | —— |
| 27 | $C_9$: Revenue distribution | 0 | 0 | 0.074 | 0.528 | 0.398 | —— | —— | —— | —— | —— |
| 28 | $C_{10}$: Operational stability | 0 | 0.009 | 0.139 | 0.509 | 0.43 | —— | —— | —— | —— | —— |
| 29 | $C_{11}$: Project Feasibility Study Report | 0 | 0.019 | 0.176 | 0.472 | 0.333 | —— | —— | —— | —— | —— |
| 30 | $C_{12}$: Cost-benefit assessment | 0 | 0.019 | 0.074 | 0.519 | 0.389 | —— | —— | —— | —— | —— |
| 31 | $C_{13}$: Performance Evaluation | 0.009 | 0 | 0.139 | 0.472 | 0.38 | —— | —— | —— | —— | —— |
| 32 | $C_{14}$: Exit mechanism | 0 | 0.019 | 0.167 | 0.5 | 0.315 | —— | —— | —— | —— | —— |

In this expression. $t_i$ is the fuzzy set of the membership, and "·" is the fuzzy operator. For example, we can calculate the membership of the external environment:

$$T_B = (0.135\ 0.121\ 0.127\ 0.125\ 0.134\ 0.113\ 0.127\ 0.118) \times \begin{bmatrix} 0 & 0.009 & 0.13 & 0.370 & 0.491 \\ 0.009 & 0.037 & 0.25 & 0.454 & 0.25 \\ 0 & 0.019 & 0.18 & 0.519 & 0.287 \\ 0 & 0.019 & 0.212 & 0.481 & 0.287 \\ 0 & 0.009 & 0.111 & 0.444 & 0.435 \\ 0.019 & 0.046 & 0.361 & 0.417 & 0.157 \\ 0 & 0.009 & 0.148 & 0.574 & 0.269 \\ 0 & 0.028 & 0.306 & 0.519 & 0.148 \end{bmatrix}$$

$$= (0.0032\ 0.02133\ 0.2073\ 0.4717\ 0.2965)$$

Step 7: Normalize the final FSE evaluation matrix and calculate a PSI for a particular factor component using the following equation:

$$PSI = \sum_{i=1}^{5} T \times L \tag{3}$$

From (3), we have

$$PSI_B = (0.0032\ 0.02133\ 0.2073\ 0.4717\ 0.2965) \times \begin{pmatrix} 1 \\ 2 \\ 3 \\ 4 \\ 5 \end{pmatrix} = 4.0368$$

Based on Step 6, we obtain the PSI of stakeholders' relationship and project management of Special Purpose Vehicle in Table 8.

**Table 8.** PSI index order.

| No. | Success Factor Group | PSI Index | Coefficients | Rank |
|-----|---------------------|-----------|--------------|------|
| 1 | Stakeholder relationships | 4.259 | 0.341 | 1 |
| 2 | External environment | 4.037 | 0.323 | 3 |
| 3 | Project management of the Special Purpose Vehicle | 4.188 | 0.336 | 2 |

The project success index for PPP projects in China is therefore expressed using the following equation:

$$\begin{aligned} PSI = \quad & 0.341 \times stakeholders\ relationship \\ & +0.323 \times external\ environmental \\ & +0.336 \times project\ management\ of\ SPV \end{aligned} \tag{4}$$

**4. Results**

Equation (4) shows that stakeholder's relationships yielded the highest coefficient (0.341) in the evaluation model, followed by project management of a Special Purpose Vehicle (0.336) and external environment (0.323). The sum of all these coefficients is one, which fits within the unity threshold. This success index equation should significantly enable practitioners in China to evaluate the success level of their PPP projects in a practical and reliable manner. What is more, the evaluation model makes it possible for practitioners to compare the success levels of two or more projects at the same level. The application of this research should improve the implementation practices of PPP projects in China.

This section discusses the top nine critical success factors that has divide into three success groupings in formulating sustainable PPP. The top three factors concerning stakeholder relationships include private sector financing capacity, government credit, and government commitment or guarantee. The top two factors related to the external environment include: completeness of legal framework and available financial markets. The top four factors related to the project management of the social purpose vehicle included: the feasibility study report and implementation, effectiveness of risk management, project investment, and cost control and revenue distribution. The high overall confirmed that the PSI was necessary for PPP projects in China.

*4.1. Stakeholder Relationships*

The stakeholder relationship category had a PFI of 4.259 and a coefficient value of 0.341 in the critical success factors evaluation model. Previous studies have also noted the stakeholder relationship category as critical criteria for most traditional construction projects [91–94].

Among the 32 critical success factors, "private sector financing capacity" was ranked at the top, mainly attributed to the reduction in the financial burden on the government. The availability of flexible and attractive financial instruments is expected important to enable the private sector to finance PPP projects; these instruments include debt, equity, supplier and purchaser credit, and securities [64]. PPP projects are generally large infrastructure construction projects, and face a paradox due to uncertainty and the fact that available information is not aligned throughout the PPP projects' life cycle [95]. Additionally, PPP projects are funded by private financing; the public sector self-finances a certain proportion of the expenses. Self-financing for the public sector and private financing require significant synergies that can contribute to PPP project success.

Government credit was the second most important factor impacting PPP project success. A failure by public agencies to fulfill their obligations in the concession contract can directly or indirectly negatively affect the project. Government credit poses a critical risk to PPP projects in different sectors [96]. A perfect credit system could improve the

efficiency of PPP implementation [97]. However, it has been reported that the probability of local public agencies breaching contracts has been relatively high in China [98,99]. In infrastructure PPP projects, good government credit is a critical factor impacting PPP project success [100]. There some PPP projects were not successful, such as failed cases 14, 17, and 19 in Table 2.

Government commitment or guarantee was ranked as the fifth most significant factor, and was attributed to improvements in the investment motivation of the private consortiums in PPP projects [101], and can influence the magnitude of political and regulatory risks [102]. Government guarantees include credit guarantees, material supply and price guarantees, minimum income guarantees, and guarantees related to exchange rates, interest rates, and inflation. PPP projects with government guarantees can maximize social-economic net present value and simultaneously optimize welfare [38]. However, a stable long-term plan for PPP projects requires enhanced certainty with respect to the government commitment or guarantee. For example, Treasury (2012) [103] launched PF2 (the latest version of PFI), which devoted a full chapter titled "Strengthening the Procurement Process." This chapter stipulates the government's commitment to 'ensuring that PF2 procurement is faster and cheaper than PFI procurement has been in the past, without sacrificing quality and competitiveness' (HM Treasury 2012 [103]). Meanwhile, government guarantees tend to stimulate an express expansion of PPP projects (MoF-China 2014 [104]); to this end, China's Ministry of Finance and the National Development and Reform Commission has promulgated a series of PPP policies since 2014.

### 4.2. External Environmental

Completeness of the legal framework was ranked sixth in importance, because of the immature legal systems in China [9]. The scholar and the practitioner have been aware of the urgent need for the Chinese government to establish a sound legal and institutional system to successfully apply PPP projects in China [105]. Meanwhile, an increasing number of renegotiations [22,51], contract variations, adjustments and arbitration [34], and early terminations [21,27,106] have already been reported by PPP project practices in China. Inadequate legal systems have been named as a critical factor restrict the development of PPP projects in China.

Available financial markets were ranked as the ninth most critical success factor for PPP projects. Many researchers [16,17,39,48,73] have found that project financing is a critical factor for private sector investment in PPP projects. The validity of an efficient and mature financial market, with the benefits of low financing costs and a diversified range of financial products would incentivize private sector pick-up of PPP projects. An unattractive financial market can create an obstacle to the implementation of PPP projects [15].

### 4.3. Project Management of Special Purpose Vehicle

Feasibility study completion and implementation planning was ranked as the third most important factor. The feasibility study provides project data and instruments that facilitate a profound analysis and that assist the PPP project's decision making process. Generally, the feasibility study is an appropriate means to illustrate the PPP project's practicability and operability. The implementation plan and data are extracted from the feasibility study for a PPP project [107]. In the life cycle of a PPP project, identifying an uncertain factor could be quite difficult, unless detailed feasibility studies have been done to assure the project's viability and enforceability [44], and it can easily lead to project failure.

Effective risk management was ranked as the fourth most important factor. PPP project risk management practices are highly variable, intuitive, subjective, and unsophisticated [108]; this is likely to lead to project failure. Many studies [109–112] have shown that risk management is a critical concern in PPP projects and the efficient allocation of risk remains problematic [113–115]. Furthermore, previous studies on PPP practices [44,89,116,117] have documented the prevalence of inefficiencies in risk allocation.

There is a clear understanding and recognition that the nature of PPP risk misallocation must be quantitatively represented and verified when investing in a life-cycle PPP project. In addition, it is necessary to balance and share the considerations needed for effective risk distribution between the public and private sectors. However, parties that facilitate project success are guided by the premise of the basic principles of sound risk management.

Project investment and cost control was ranked as the seventh most critical factor. PPP project investments depend heavily upon private capital markets for financing and depend on private firms for managerial expertise. Since 2013, when China's economic growth entered a transition, the risk of debt exposure emerged, and PPP projects became a main approach for infrastructure construction. Since China's promotion of PPP projects, PPP project investments have gotten out of control, leading to an increase in government expenditure responsibility. This directly affects earnings to public and private sector.

Revenue distribution was listed as the ninth most critical factor. When revenue is distributed, the two parties compete for interests and strive to minimize their own opportunity costs. Under market competition, public and private sector achieve a win–win situation through cooperation and competition [118]. Nonetheless, unreasonable revenue distribution can affect a project's normal operation. There is the need for a revenue distribution mechanism, where the government ensures extra revenues. Therefore, identifying revenue as an attribution mechanism is indispensable as a suitable mitigation strategy to mitigate traffic revenue risks in PPP transport infrastructure projects [112].

## 5. Conclusions

PPP projects have been implemented to support infrastructure development in both developed and developing countries with diverse results, and many researchers claimed that PPP can contribute to sustainability in China as it promotes long-term productive use of resources [119,120]. These provide a mechanism for investing in public infrastructure, while also effectively transferring the government function to the private sector. Meanwhile, this generates significant problems as an increasing number of project failures appear. In China, from 2013 to 2019 (years inclusive), CPPPC data (http://www.cpppc.org:8086 /pppcentral/map/toPPPMap.do, accessed on 30 December 2020) show that the market capacity for PPP projects reached nearly 10,000 projects, with a total investment of more than 13.7 trillion yuan BRI data (http://www.bridata.com/, accessed on 30 December 2020) show that China's PPP projects occupy a market share of 15.4 trillion yuan, with the number of PPP projects reaching 10,226 projects. However, with the release of normative documents from central government in China, thousands of PPP projects have been withdrawn from the CPPPC library in the large PPP market. Those unreasonable exit phenomenon needs are more detailed identification of critical success factors for PPP projects. Then, this study defined and categorized the factors affecting project success and failure. From this classification and definition, we applied a fuzzy synthetic method to prioritize these factors and provide an evaluation criterion.

In fact, by using the fuzzy synthetic evaluation model for PPP projects, the most critical success factors for different types of PPP projects could be identified and both precautionary and remedial actions could be taken as soon as possible. Both the public sector and private sector can adopt this model to assess the risk level of their PPP projects. And the results can be used to compare the critical success factor levels with their counterparts for benchmarking purposes. Such an extension would provide a deeper understanding of managing different types of PPP projects. Since the critical success factor level may vary at various stages of a project life cycle, it is worthwhile to develop a PPP fuzzy synthetic evaluation model for measuring critical success factors across different stages of a project life cycle in future.

First, due to the wide range of success factors and categories amassed by researchers [121], this study reviewed recent literature and cases to define the success factors of PPP projects during the period 2000 to 2019, highlighting the research contributions by various countries with respect to their authors and institutions. Therefore, 32 success factors were sorted from

recent literature and cases. These were then divided into three dimensions: stakeholder relationships, external environment, and the project management of a special purpose vehicle (SPV).

Second, based on 32 defined factors, a questionnaire was designed and distributed to experts, researchers, and PPP project managers. Survey data were then collected, and mean score values of the response data were used to rank the relative importance of 32 critical success factors in the China PPP environment. Then, 10 factors emerged as being most important in developing a successful China PPP: private sector financing capacity, government credit, feasibility study, effective risk management, government commitment, completeness of legal framework, project investment control, revenue distribution, available financial markets, and operational feasibility.

Finally, a fuzzy synthetic method was applied to prioritize the critical success factors. Despite the model's applications and the survey and case study results, this research did have some constraints. Extending the sample frame to other type of PPP projects could improve the validity of the research model. Examining a similar model with other projects and other countries and comparing them could yield practical results

Different success factors were identified using a questionnaire survey, case studies, literature review, fuzzy synthetic methods, and interviews and correspondence with worldwide PPP experts and practitioners. Furthermore, this assessment provides results in terms of the performance of dominant CSFs. This can be useful when prioritizing PPP project tasks. These approaches are valid, could be used globally for other PPP projects, and may also be evaluated with respect to CSFs in a PPP context.

**Author Contributions:** B.D., D.Z. and Y.Y., conceived the study; D.Z., J.Z. and X.L. conducted the literature review, developed the model, designed the experiments and performed the experiments; B.D., D.Z. and J.Z. analyzed the data and results; B.D., D.Z., J.Z. and X.L. edited the final version of the paper. All authors have read and agreed to the published version of the manuscript.

**Funding:** 1. the National Natural Science Foundation of China: 71602144; 2. the Tianjin education commission Project of Key Research Institute of Humanities and Social Sciences at Universities: 2017JWZD15; 3. the Program for Innovation Research Team in Universities of Tianjin: TD13-5019.

**Institutional Review Board Statement:** Not applicable.

**Informed Consent Statement:** Not applicable.

**Data Availability Statement:** Not applicable.

**Acknowledgments:** The authors thank the National Natural Science Foundation of China (Grant No. 71602144), the Tianjin education commission Project of Key Research Institute of Humanities and Social Sciences at Universities (Grant No. 2017JWZD15), and the Program for Innovation Research Team In Universities of Tianjin (TD13-5019).

**Conflicts of Interest:** The authors declare that they have no competing interests. Authors' contributions: all authors contributed equally and significantly in writing this paper. All authors read and approved the final manuscript.

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
