# Peer review of "Fuzzy Synthetic Evaluation of the Critical Success Factors for the Sustainability of Public Private Partnership Projects in China"

_sustainability, doi:10.3390/su13052551_

Round 1
Reviewer 1 Report
Some suggestions to improve the quality of the article:
- Row 70: can you provide statistics, not only references, regarding the importance (rate) of the factors that led to project failures? And please explain how did they affect the projects` success.
- The Literature Review shows the actual state of the art. It`s the author`s base for finding gaps and fill them. Which are these gaps, as long as previous studies listed all the factors found in the your study? Please explain.
- Table 5: are the subjects representative for your study? Which is the rate of private sector, public sector, or research institutions in China`s economy? Please provide information.
- Is this study relevant and valid? I don`t know. 108 subjects, but thousands of projects every year. How many projects did the subjects develop? How many of them were not successful? How did you choose the target group?
Author Response
Thanks for reviewer comments and suggestions on our paper. The authors have learned much from it, and carefully proofread the manuscript and edit it as following:
- Row 70: can you provide statistics, not only references, regarding the importance (rate) of the factors that led to project failures? And please explain how did they affect the projects` success.
Answer:
This paper is about the statistics of the influencing factors, not only references (from Web of Science database and CNKI database in China), but also cases study were conducted out to collect data from 40 successful and failed PPP projects in China (Table 2). Following the statistics of the influencing factors by references analysis and cases study, the author could just figure out the number of occurrences of the influencing factors, not the rate (because the diversity and heterogeneity for different PPP projects). Rank the influencing factors according to the data frequency, and finally obtained the 32 CSFs in three groups (see Table 4)
- The Literature Review shows the actual state of the art. It`s the author`s base for finding gaps and fill them. Which are these gaps, as long as previous studies listed all the factors found in the your study? Please explain.
Answer:
The researcher read previous studies to ensure that invalid records were not included. Table 1 lists 30 critical success factors from the document analysis, and added the following description in this paper.
From the above-selected literature, similarities of the success factors for PPPs are obvious, and priority is placed on nominating perceived CSFs based on perception of public and private sector participants. A large proportion of the reviewed studies arrived at their nominated CSFs based on their mean scores or experience analysis[60-88]. Therefore, it is imperative to establish and statistic the key principal success factors in life cycle of PPP projects, their interrelationships, management principles and contribution to successful implementation of a candidate project.
- Table 5: are the subjects representative for your study? Which is the rate of private sector, public sector, or research institutions in China`s economy? Please provide information.
Answer:
Table 5 shows sectors and experience levels in PPP projects. A total of 49.07% of respondents were from engineering advisory institutions; 24.07% came from universities or research institutions; 18.52% came from the private sector, 2.78% came from financial institutions, and 3.7% came from other types of organizations.
- Is this study relevant and valid? I don`t know. 108 subjects, but thousands of projects every year. How many projects did the subjects develop? How many of them were not successful? How did you choose the target group?
Answer: The author included the following description in this paper.
Since respondents may be engaged in multiple types of PPP projects, in order to avoid the problem of overgeneralization, the author made multiple choice on the type of PPP projects the respondent has been engaged in questionnaire. The results showed that most of PPP projects engaged by respondents are distributed in the following figure: such as 63 transportations, 46 water conservancy, 55 ecological construction and environmental protection, 65 municipal engineering, 42 government infrastructure construction, 29 comprehensive pipeline development, etc. This data conforms to the current development trend of PPP projects. Therefore, it is crucial to identify and analyze the key success factors of PPP projects.
Figure 2 Types of PPP projects from respondents (in word file)

Reviewer 2 Report
- The research identified, classified, and evaluated the success factors that may affect PPP project for achieving sustainability. The topic is good issue in this time.
- The questionnaire survey was conducted, with 108 responses received from experts, researchers, and PPP project managers in China. How did the researchers decide the participants? Please make a clear description.
- Employing ing a fuzzy synthetic evaluation (FSE) method, stakeholder relationships (A1-A10), external environmental (B1-B8), and project management of a special purpose vehicle (C1-C14) collected at three different factor group located in PPP projects was used in this evaluation. Please explain the three different factor groups in this study.
- The study was demonstrated that fuzzy synthetic evaluation techniques are quite appropriate techniques for PPP projects. How could this study approve this viewpoint?
- The research findings should impact on policy development towards PPP and Private Finance Initiative (PFI) project governance. How could the study affect project governance? What is the effeteness?
Author Response
Thanks for reviewer comments and suggestions on our paper. The authors have learned much from it, and carefully proofread the manuscript and edit it as following:
The research identified, classified, and evaluated the success factors that may affect PPP project for achieving sustainability. The topic is good issue in this time.
- The questionnaire survey was conducted, with 108 responses received from experts, researchers, and PPP project managers in China. How did the researchers decide the participants? Please make a clear description.
Answer: Line 233, in Table 5, the author received 108 valid questionnaires from experts. Thus, private sectors of experts are 20, financial institutions of experts are 3, financial institutions of experts are 53, universities or research institutions of experts are 26, public sectors of experts are 2, others experts are 4.
- Employing a fuzzy synthetic evaluation (FSE) method, stakeholder relationships (A1-A10), external environmental (B1-B8), and project management of a special purpose vehicle (C1-C14) collected at three different factor group located in PPP projects was used in this evaluation. Please explain the three different factor groups in this study.
Answer: Based on a literature review (Table 1, P4-P6) and case summaries (Table 2, P7, and Table 3 P8), this study divided project success factors into three dimensions: relationships between stakeholders, project management in a Special Purpose Vehicle (SPV), and the external environment of a PPP project. First, the relationship between stakeholders included each participant’s behavior, and the partnership and the contractual relationship, including the technical ability of the social sector, government credit, and other factors. Second, the project management of a SPV is composed of technology and management factors, impacting the project success in project life cycle management. These examples include risk allocation in risk management, investment control, and other factors. Third, the external environmental holds un-controllable factors that affect the implementation effect of PPP projects, such as a sound legal framework and credible economic policies. Therefore, after collection, screening, and analysis processes, the literature research and case analysis yielded a final list of 32 CSFs (named Ai, Bi, or Ci) and grouped as A, B, and C on Table 4 (P9).
- The study was demonstrated that fuzzy synthetic evaluation techniques are quite appropriate techniques for PPP projects. How could this study approve this viewpoint?
Answer: In fact, by using the fuzzy synthetic evaluation model for PPP projects, the most critical success factors for different types of PPP projects could be identified, and both precautionary and remedial actions could be taken as soon as possible. Both the public sector and the private sector can adopt this model to assess the risk level of their PPP projects. And the results can be used to compare the critical success factor levels with their counterparts for benchmarking purposes. Such an extension would provide a deeper understanding of managing different types of PPP projects. Since the critical success factor level may vary at various stages of a project life cycle, it is worthwhile to develop a PPP fuzzy synthetic evaluation model for measuring critical success factors across different stages of a project life cycle in future.
- The research findings should impact on policy development towards PPP and Private Finance Initiative (PFI) project governance. How could the study affect project governance? What is the effeteness?
Answer: This research investigates the effects of critical success and failure factors through the effectiveness of the fuzzy synthetic evaluation model for PPP projects. The aim of this research can produce a list critical success factors (GSFs) to reduce the failure rate of PPP projects in future. And these critical success factors can be used in the future research to build the project governance framework for PPP projects. In future, the project governance framework will be made up of a list of 32 critical success factors that were categorized into 3 groups, and proposed to the project sector or private sector to apply on PPP projects for evaluating the significance.

Reviewer 3 Report
Dear authors,
As they indicate, the study has its limitations (lines 501-505), as it is limited to a special purpose vehicle (SPV). Is the case study also limited to this type of project? And the bibliographic review? This information is important, since the first mention of the SPV is in line 186, when the three dimensions are proposed.
On the other hand, I would also recommend a greater justification in the classification of the three proposed dimensions (Stakeholder relationships, External environmental and Project management of a Special Purpose Vehicle (SPV), mentioned in lines 183-187, they would allow to finish correctly supporting the research, and subsequent data obtained in the analysis.
Finally, in the conclusions, the results obtained with the bibliographic review, which is quite extensive in the first sections, should also be supported.
Best regards.
Author Response
Response to reviewer 3
Thanks for reviewer comments and suggestions on our paper. The authors have learned much from it, and carefully proofread the manuscript and edit it as following:
As they indicate, the study has its limitations (lines 501-505), as it is limited to a special purpose vehicle (SPV). Is the case study also limited to this type of project? And the bibliographic review? This information is important, since the first mention of the SPV is in line 186, when the three dimensions are proposed. On the other hand, I would also recommend a greater justification in the classification of the three proposed dimensions (Stakeholder relationships, External environmental and Project management of a Special Purpose Vehicle (SPV), mentioned in lines 183-187, they would allow to finish correctly supporting the research, and subsequent data obtained in the analysis. Finally, in the conclusions, the results obtained with the bibliographic review, which is quite extensive in the first sections, should also be supported.
Answer: Thanks to the reviewer's suggestions, the author made the following improvements:
Firstly, after 2013, numerous PPP projects have been implemented using a Special Purpose Vehicle (SPV) type of enterprises under the policy of China (PPP Project Operation Manual, 2015, http://www.ccgp.gov.cn/ppp/llyj/201512/t20151215_6324231.htm), those involves all cases study and most of bibliographic review. According to the proportion of equity that stipulated in PPP contracts, public sector and private sector should contribute to establish SPV. It is essentially a partnership between public sector organizations and private sector investors primarily in the form of a standalone business venture. The main task of the SPV is to implement the project until it is put into operation. Thus, operating and managing relationships for SPV, this paper incorporated SPV in three-dimensional analysis framework.
Secondly, 32 factors is divided into three groups for PPP projects which, if rightly put together and provided special and sustained management attention, would enhance the likelihood of successful implementation of such projects in China. In other words, these factors have benefit influence on implementation of the PPP project.
At last, in the conclusions, the author modified the parts of results, which can response the bibliographic review.

Reviewer 4 Report
The paper is interesting. The following constructive comments should be addressed before the final review and acceptance: Clarify the research question in Section 1, and discuss the contributions of the work. Table 1: too many references are listed here without mentioning what the say collectively and what they say individually. It is like mass citation without any value for readers. Line 499: why “a fuzzy synthetic method” adds value to this investigation? Why fuzzy is required for factor identification. In the conclusion, insert the limitations and future studies. The practical implications and theoretical implication of the study (e.g. factors identified) should be clarified here. Line 2018: how it is distributed? I don’t mean email, I mean what population and how the sample is taken? How did you collected their email address? How randomly it is and discuss if the outcome can be generalized to the entire population? Insert a discussion in terms of sample size. Mention that the entire process of data collection and analysis is reliable and valid. The reviewer suggests to read and cite this discussion since it refers to some items that should be considered in conducting surveys: 2016. Discussion of “Barriers of Implementing Modern Methods of Construction” by M. Motiar Rahman. Journal of Management in Engineering, 32(2), p.07015001. https://doi.org/10.1061/(ASCE)ME.1943-5479.0000410 Can you statistically discuss how significant is the different between the following groups of respondents? Private vs other sectors Less experienced vs experienced participants. Thanks for drafting the paper.Author Response
Thanks for reviewer comments and suggestions on our paper. The authors have learned much from it, and carefully proofread the manuscript and edit it as following:
The paper is interesting. The following constructive comments should be addressed before the final review and acceptance:
- Clarify the research question in Section 1, and discuss the contributions of the work.
Answer:
Inspired by the above literature and research, this study prioritized the factors significantly influencing PPP projects. This included applying a fuzzy synthetic evaluation analysis method to overcome the issues of interdependencies and feedback among different factor-ranking alternatives. This research also developed a checklist of CSFs for PPP, which could be adopted in the further empirical and sustainable research.
Using the fuzzy synthetic evaluation method, the main aim of this paper to cover the necessity of study critical success factors, and that are obtained top nine factors greatly impact on the success of PPP projects, such as private sector financing capacity, government credit, feasibility study, effective of risk management, government commitment, completeness for legal framework, project Investment control, revenue distribution, available financial markets, feasibility of operation, ate. Although the ranking of many factors were different between public and private sectors, there were no significant differences in the perception of public and private sectors concerning the importance of the success factors except for a few factors. This paper highlights not only the important success factors for PPP implementation in China, but also offers evidence concerning the importance of the factors of the public sector and the private sector involved in PPP projects.
- Table 1: too many references are listed here without mentioning what the say collectively and what they say individually. It is like mass citation without any value for readers.
Answer: The systematic research PPP projects, “critical success factor” and “PPP project” were utilized as search keywords to identify journal papers published from 2000 to 2019 in international journals using the China National Knowledge Infrastructure (CNKI) database in China and Web of Science database. The author obtained 279 papers after the data-cleaning process, including 186 Chinese papers and 93 international journal papers. The researcher read these papers to identify various success factors that are further analyzed, distilled, coded, and classed into 3 groups. Thus, combined typical case studies of PPP projects in China, and using the fuzzy synthetic evaluation method, the main aimt of this paper to cover the necessity of study critical success factors, and that are obtained top nine factors greatly impact on the success of PPP projects.
- Line 499: why “a fuzzy synthetic method” adds value to this investigation? Why fuzzy is required for factor identification.
Answer: Using the fuzzy synthetic method, the extracted CSFs were used to construct a predictive tool for assessing the likelihood of a successful project implementation. These factors are (in order): private sector financing capacity, government credit, feasibility study, effective of risk management, government commitment, completeness for legal framework, project investment control, revenue distribution and available financial markets.
- In the conclusion, insert the limitations and future studie”s. The practical implications and theoretical implication of the study (e.g. factors identified) should be clarified here.
Answer: Thanks for the reviewer's suggestion, the author will add the following paragraph of limitations and future studie’s for this paper in the conclusion.
This study has not provided a complete list of all the possible success factors that might impact on the implementation of PPP projects. However, the convergence with the literature provides confidence in the findings, and for a specific project in a country, there would be unique factors that should be added fuzzy synthetic method to identified critical success factors. Future research could adopt a more locally focused interviews and case study analysis to unravel critical success factors in managing operational PPPs. Nonetheless, the adopted checklist of CSFs, such as private sector financing capacity, government credit, feasibility study, effective of risk management, government commitment, completeness for legal framework, project Investment control, revenue distribution, available financial markets and feasibility of operation related to operational PPPs. Thus, this work provides direction for both future research and future China event practice, based on the large-sample, the quantitative finding related to the most critical success factors for China activities and the development of supporting fuzzy synthetic method.
- Line 218: how it is distributed? I don’t mean email, I mean what population and how the sample is taken? How did you collected their email address? How randomly it is and discuss if the outcome can be generalized to the entire population? Insert a discussion in terms of sample size. Mention that the entire process of data collection and analysis is reliable and valid.
Answer:
The questionnaire survey was sent by email, because of the corresponding author's institution that is a executive director unit of “PPP Forum of Chinese Universities”. It also hosted five China PPP Forums, since 2016 to 2020. According to email in forum address book, the author contacted some expert, they working in private sectors, financial institutions, advisory institutions, universities or research institutions, public sectors, ate. However, despite the methodological limitations of the small number of responses, small samples are not rare in an international e-mail/web-survey based research in PPP studies (Ameyaw and Chan, 2015a, Osei-Kyei and Chan, 2016, 2017).
Albert, P.C, Chan,et al.. Risk ranking and analysis in PPP water supply infrastructure projects: An international survey of industry experts[J]. Facilities, 2015.
Robert, Osei, -, et al. Empirical comparison of critical success factors for public-private partnerships in developing and developed countries: a case of Ghana and Hong Kong[J]. Engineering Construction & Architectural Management, 2016.
Robert O K , Chan A P C . Implementing public–private partnership (PPP) policy for public construction projects in Ghana: critical success factors and policy implications[J]. International Journal of Construction Management, 2017, 17(2):113-123..
- The reviewer suggests to read and cite this discussion since it refers to some items that should be considered in conducting surveys: 2016. Discussion of “Barriers of Implementing Modern Methods of Construction” by M. Motiar Rahman. Journal of Management in Engineering, 32(2), p.07015001. https://doi.org/10.1061/(ASCE)ME.1943-5479.0000410
Answer: The author read and cited the following literature in this paper, thanks reviewer suggestion.
Rahman M.M. Barriers of Implementing Modern Methods of Construction [J]. Journal of Management in Engineering, 2013, 30(1):69-77.
- Can you statistically discuss how significant is the different between the following groups of respondents?
Answer: The experts were selected based on these pre-defined criteria. The respondent should have extensive working or research experience in PPP project delivery, the respondent should also have in depth knowledge or practical experience on the operational management of PPP projects ,and lastly the respondent should have in-depth knowledge on PPP project success.
Respondents represented private sectors, financial institutions, advisory institution universities, research institutions, and public sectors. Table 5 shows the sectors and the experience levels in PPP projects. A total of 49.07% of respondents were from engineering advisory institutions; 24.07% came from universities or research institutions; 18.52% came from the private sector, 2.78% came from financial institutions, and 3.7% came from other types of organizations. The distribution of respondents was consistent with PPP project stakeholders, essentially representing all stakeholders across the PPP project life cycle.

Round 2
Reviewer 1 Report
Much better version of the article.
However, I don`t think that 108 subjects are relevant for the study.
Author Response
Thank you very much for your attention and the referee’s evaluation and comments on our paper. The authors have revised and interpreted the manuscript according to your kind advices and detailed suggestions.
There are certain differences in the results of the questionnaire survey from various groups of respondents (Showed in Table 5). In fact, the data about the respondents' number of working years were as follows: 31.48% had less than 2 years of work experience; 50% had 2-5 years of experience; and the others had more than 5 years of experience. Among the 108 questionnaires managed by the respondents, 68.52% had more than 5 years of working years, with rich work experience. This screening information ensured quality, reduced the occurrence of potential risks, and improved the accuracy of the research conclusions.
In addition, the respondents has engaged in more than three types of PPP projects (such as transportations, water conservancy, ecological construction and environmental protection, municipal engineering, government infrastructure construction, comprehensive pipeline development, etc.), because of the respondents from different sectors (such as Private sector, financial institution, advisory institution, universities or research institutions, public sector, etc.). Based on the feedback data of the questionnaire, the author believe it authenticity, although the size of research samples are relatively small.
Although the size of research samples are relatively small, it also took more than half a year of data collection and collation by email, and finally obtained the corresponding feasibility conclusion. However, the ranking of critical success factors in the conclusion that may have some deviation before and after. In fact, it reveals the common problems in China's PPP projects, currently. what is more, the aim of this paper is helping PPP projects to set contract terms and procedures in advance, and avoid project risks as much as possible during operation.
Reviewer 4 Report
The paper improved. The following items have not properly addressed:
The question is how the results are different in various groups of respondents. Is there any statistically significant difference?
The paper by M. Motiar Rahman itself is less relevant than the “discussion” published on this paper. The discussion drafted on this paper is important since it shows many tips about issues that in a survey should be considered, particularly in the tables inserted in that paper. I would suggest mentioning this “discussion” in the method section and confirm all items relevant to a survey was considered: 2016. “Discussion of Barriers of Implementing Modern Methods of Construction” Journal of Management in Engineering, 32(2), p.07015001. https://doi.org/10.1061/(ASCE)ME.1943-5479.0000410
Thanks for addressing some comments.
Author Response
Thank you very much for your attention and the referee’s evaluation and comments on our paper. The authors have revised and interpreted the manuscript according to your kind advices and detailed suggestions.
Firstly, there are certain differences in the results of the questionnaire survey from various groups of respondents. Table 5 shows the sectors and the experience levels in PPP projects. A total of 49.07% of respondents were from engineering advisory institutions; 24.07% came from universities or research institutions; 24.07% came from universities or research institutions; 18.52% came from the private sector, 2.78% came from financial institutions, and 3.7% came from other types of organizations. In addition, screening responses to questionnaires and respondents that may be engaged in multiple types of PPP projects, in order to avoid the problem of overgeneralization, the author made multiple choice on the type of PPP projects the respondent has been engaged in questionnaire (see Figure 2). The results showed that the most of PPP projects engaged by respondents are distributed in the following that: such as 63 transportations, 46 water conservancy, 55 ecological construction and environmental protection, 65 municipal engineering, 42 government infrastructure construction, 29 comprehensive pipeline development, etc. This data conforms to the current development trend of PPP projects. Therefore, it is crucial to identify and analyze the key success factors of PPP projects.
Secondly, The author read and cited the following literature in this paper, thanks reviewer suggestion.“Rahman M.M. Barriers of Implementing Modern Methods of Construction [J]. Journal of Management in Engineering, 2013, 30(1):69-77”。
Thirdly, the authors read and cited the following literature in this paper, thanks reviewer suggestion.
Bliemel, M, Bemanian, et al. Discussion of "Barriers of Implementing Modern Methods of Construction" by M. Motiar Rahman[J]. Journal of management in engineering, 2016, 32(2):7015001.1.